# All fractional Shapiro steps in the RSJ model with two Josephson harmonics

**Pavel N. Tsarev[1,2⋆] and Yakov V. Fominov[1,2,3†]**

**1** L. D. Landau Institute for Theoretical Physics RAS, 142432 Chernogolovka, Russia
**2** Moscow Institute of Physics and Technology, 141701 Dolgoprudny, Russia
**3** Laboratory for Condensed Matter Physics, HSE University, 101000 Moscow, Russia

⋆ tsarev.pn@phystech.edu , † fominov@itp.ac.ru

## Abstract

Synchronization between the internal dynamics of the superconducting phase in a Josephson junction (JJ) and an external ac signal is a fundamental physical phenomenon, manifesting as constant-voltage Shapiro steps in the current-voltage characteristic. Mathematically, this phase-locking effect is captured by the Resistively Shunted Junction (RSJ) model, an important example of a nonlinear dynamical system. The standard RSJ model considers an overdamped JJ with a sinusoidal (single-harmonic) current-phase relation (CPR) in the current-driven regime with a monochromatic ac component. While this model predicts only integer Shapiro steps, the inclusion of higher Josephson harmonics is known to generate fractional Shapiro steps. In this paper, we show that only two Josephson harmonics in the CPR are sufficient to produce all possible fractional Shapiro steps within the RSJ framework. Using perturbative methods, we analyze amplitudes of these fractional steps. Furthermore, by introducing a phase shift between the two Josephson harmonics, we reveal an asymmetry between positive and negative fractional steps—a signature of the Josephson diode effect.

# 1  Introduction

The Resistively Shunted Junction (RSJ) model [1–3], originally derived to describe the Josephson effect in superconductors, has emerged as a paradigmatic example of a nonlinear dynamical system with interdisciplinary applications. In the context of overdamped Josephson junctions (JJs), its standard form is

$$\frac{\hbar}{2eR}\frac{d\varphi}{dt} + I_s(\varphi) = I_{\text{dc}} + I_{\text{ac}}\cos\omega t, \tag{1}$$

where $R$ is the normal-state junction resistance and the function sought is $\varphi(t)$. The model contains three key ingredients: (i) Dissipation ($d\varphi/dt$ term) models energy loss, crucial for realistic physical systems. (ii) Periodic force [$2\pi$-periodic $I_s(\varphi)$ term] reflects phase coherence in superconductors and may arise due internal periodic potential in other systems. (iii) Driving force [the right-hand side (rhs) of Eq. (1)] represents external control parameters with monochromatic ac component (dc/ac currents, forces, fields).

This minimalistic yet rich structure allows the RSJ equations to describe various physical phenomena such as the Josephson effect in superconductors [4–7], the charge transport by charge density waves in Peierls conductors [8–10], motion of mechanical systems such as an overdamped pendulum (mechanical analog of the JJ) [11] or a Suslov system (specific type of a rigid-body motion) [12], etc. Mathematically, the RSJ model is studied as an example of nonlinear dynamics [13–19].

In the absence of an external ac driving component, i.e., at $I_{\text{ac}} = 0$, the solution of Eq. (1) depends on relation between $I_{\text{dc}}$ and the critical current $I_c$ defined as the maximum of the current-phase relation (CPR) $I_s(\varphi)$. At $I_{\text{dc}} < I_c$, the system is in the stationary state with phase $\varphi$ corresponding to the fixed external current $I_{\text{ac}}$. At $I_{\text{dc}} > I_c$, the system enters the running state with time-dependent phase $\varphi(t)$. In this resistive regime, a finite voltage

$$V(t) = \frac{\hbar}{2e}\frac{d\varphi}{dt} \tag{2}$$

across the junction appears, and its time average defines the current-voltage characteristic (CVC) $\overline{V}(I_{\text{dc}})$. The solution of Eq. (1) demonstrates Josephson oscillations corresponding to winding of $\varphi(t)$ by integer multiples of $2\pi$.

In the presence of external ac driving, the internal Josephson oscillations can synchronize with external driving. This leads to phase locking, manifesting as Shapiro steps in the CVC [4,5]. Generally, these constant-voltage steps appear at

$$\overline{V} = \pm\left(\frac{n}{k}\right)\frac{\hbar\omega}{2e}, \tag{3}$$

with natural numbers $n$ and $k$.

The simplest current-phase relation corresponds to sinusoidal CPR, $I_s(\varphi) = I_c\sin\varphi$, which can be experimentally realized in the case of tunnel Josephson junction (or in the vicinity of

the superconducting critical temperature). In this case, only integer Shapiro steps arise (corresponding to $k = 1$) [6, 7, 20–22]. Fundamental importance of the phase-locking effect in this limit is underlined by applications of tunnel Josephson junctions for the fundamental-constant measurement and the quantum metrology as a whole [6, 7]. In particular, measurements of the Shapiro steps provide the basis of the voltage standard.

In Josephson junctions with higher transparency, higher Josephson harmonics become essential [6, 7, 23], so that the CPR can be represented as

$$I_s(\varphi) = \sum_{m=1}^{\infty} I_m \sin m\varphi. \tag{4}$$

The fractional (subharmonic) Shapiro steps with $k > 1$ are then expected to appear, and their positions and amplitudes provide information on the CPR [23]. Is there direct correspondence between the fractionality of the Shapiro steps (values of $k$) and the Josephson harmonics in the CPR Eq. (4) (values of $m$)? In other words, do irreducible Shapiro fractions with a given $k$ appear only due to the presence of the harmonics with numbers $m$ that are multiples of $k$ in the CPR [24, 25]? It turns out that the answer is negative, and already two harmonics are sufficient to generate subharmonic Shapiro steps of arbitrary fractionality.

Specifically, we demonstrate in the present paper that the CPR with only two Josephson harmonics,

$$I_s(\varphi) = I_1 \sin \varphi + I_1 \sin 2\varphi, \tag{5}$$

actually generates all fractional Shapiro steps. The CVC then has a Cantor-ladder-type form (also known as the devil's staircase). We analytically find the steps' amplitudes in limiting cases.

## 2 General equations

We will investigate Shapiro steps in the framework of the RSJ model in the current-driven regime, see Eq. (1). We assume that due to external irradiation, the current has an ac component with frequency $\omega$ which sets natural units of time, voltage, and current. The corresponding dimensionless quantities are

$$\tau = \omega t, \quad \text{v} = V/V_0, \quad j_\alpha = I_\alpha/I_0 \tag{6}$$

(with $\alpha = s, \text{dc}, \text{ac}, 1, 2$), respectively, where

$$V_0 = \hbar\omega/2e, \quad I_0 = V_0/R. \tag{7}$$

In these notation, the RSJ equations (1)-(2) take the following form:

$$\text{v} = \dot{\varphi}, \tag{8}$$

$$\dot{\varphi} + j_s(\varphi) = j_{\text{dc}} + j_{\text{ac}} \cos(\tau + \beta), \tag{9}$$

where the dot denotes $d/d\tau$. In comparison to Eq. (1), we introduce the $\beta$ phase of the ac current. Obviously, it can be eliminated by a time shift; however, we prefer to keep it in order to trace the relative phase between the internal Josephson oscillations and the external ac signal.

The CVC of the Josephson junction is the $\bar{\text{v}}(j_{\text{dc}})$ dependence, where the overline denotes averaging with respect to $\tau$. Due to the ac component of the current (external irradiation), the Shapiro steps in the CVC generally appear at

$$\bar{\text{v}} = \pm n/k, \tag{10}$$

with natural numbers $n$ and $k$ [this is the dimensionless form of Eq. (3)]. Implying irreducible fractions, we refer to the steps with $k = 1$ and $k > 1$ as integer and fractional ones, respectively.

For future reference, we consider the textbook case in which the current has only the dc component and the CPR contains only the first Josephson harmonic. Equation (9) then takes the form

$$\dot\varphi + j_1 \sin\varphi = j_{dc}, \tag{11}$$

and in the case $|j_{dc}| > j_1$, its solution is [2, 3, 26]

$$\varphi(\tau) = 2\arctan\left[\frac{j_1 + v\tan(v\tau/2)}{j_{dc}}\right] + 2\pi n, \qquad v(\tau) = \frac{j_{dc}^2 - j_1^2}{j_{dc} + j_1\cos(v\tau - \alpha)}, \tag{12}$$

where

$$v = \mathrm{sgn}(j_{dc})\sqrt{j_{dc}^2 - j_1^2}, \quad \cos\alpha = j_1/j_{dc}, \quad \sin\alpha = \sqrt{j_{dc}^2 - j_1^2}/|j_{dc}|. \tag{13}$$

In this case, the CVC takes the form

$$\bar{v} = v. \tag{14}$$

Adding the ac component of the current to the rhs of Eq. (11) would lead to the appearance of integer Shapiro steps in the CVC.

# 3 Two harmonics in the current-phase relation

Our main focus is on the case of a Josephson junction with two harmonics in the CPR, see Eq. (5). We are interested in studying the fractional Shapiro steps in this case. Substituting the corresponding dimensionless $j_s(\varphi)$ into Eq. (9), we obtain the resulting differential equation

$$\dot\varphi + j_1 \sin\varphi + j_2 \sin 2\varphi = j_{dc} + j_{ac}\cos(\tau + \beta), \tag{15}$$

which we deal with below. For future use, we define $A = j_2/j_1$ as the amplitude ratio of two harmonics.

It is well-known that the Josephson junction with two harmonics in the CPR produces integer ($k = 1$) and half-integer ($k = 2$) Shapiro steps [27]. At the same time, numerical solution of Eq. (15) demonstrates the presence of all fractional steps (with arbitrary $k$), see Fig. 1(a). For this reason, we refer to the integer and half-integer steps as trivial ones, while the fractional Shapiro steps with $k \geq 3$ we call nontrivial.

## 3.1 Limit of weak external irradiation and small second harmonic ($j_{ac} \ll j_s$, $A \ll 1$)

To begin with, we consider the limit in which the ac component of the current is small compared to the maximum of $j_s(\varphi)$ (which we briefly denote as $j_{ac} \ll j_s$). In order to calculate the amplitudes of the Shapiro steps, we employ the perturbation theory with feedback [28, 29], which is well suited to treat resonances. In Eq. (15), we represent $\varphi$ and $j_{dc}$ as series with respect to small $j_{ac}/j_s$,

$$\varphi(\tau) = \varphi_0(\tau) + \varphi_1(\tau) + \varphi_2(\tau) + \ldots, \tag{16}$$

$$j_{dc} = j_{dc}^{(0)} + j_{dc}^{(1)} + j_{dc}^{(2)} + \ldots \tag{17}$$

The essence of the perturbation theory with feedback is that at each step we adjust the value of $j_{dc}$ [in accordance with Eq. (17)] in order to avoid divergence of $\varphi_n(\tau)$ at large $\tau$.

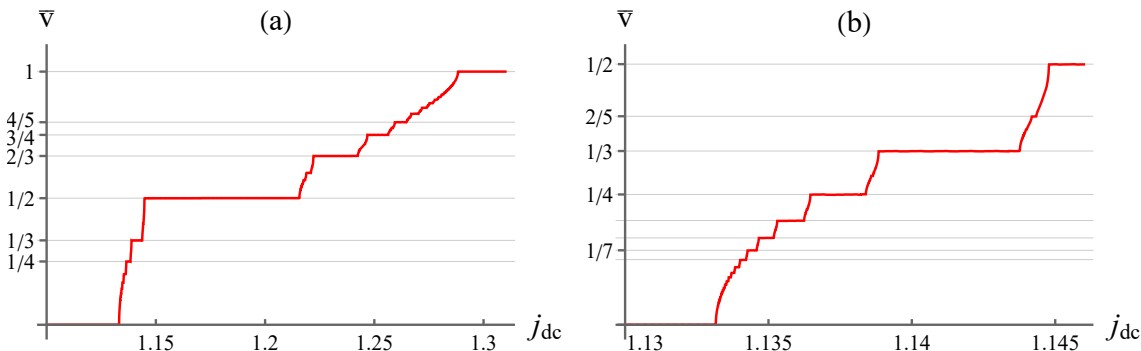

Figure 1: CVC of the Josephson junction with second Josephson harmonic in the CPR at $j_{ac} = 0.8$, $j_1 = 1$, and $A = 0.7$. (a) Range of voltages below the first Shapiro step. Many fractional steps besides half-integer one are seen. (b) Range of voltages below the $1/2$ Shapiro step [zoomed part of plot (a)]. Series of steps of type $1/k$ with $k > 1$ and the $2/5$ step, which is part of $2/k$ series are seen. Amplitude of the $2/5$ step is much less than amplitude of the steps of type $1/k$.

The zeroth approximation $\varphi_0$ is a solution of equation

$$\dot{\varphi}_0 + j_1 \sin \varphi_0 + j_2 \sin 2\varphi_0 = j_{dc}^{(0)}, \tag{18}$$

while the corrections to the phase are written as

$$\varphi_n(\tau) = \dot{\varphi}_0(\tau) \int_{\tau_n}^{\tau} \left[ j_{dc}^{(n)} + F_n(t) \right] \frac{dt}{\dot{\varphi}_0(t)}, \tag{19}$$

where $\tau_n$ are arbitrary constants and the first several $F_n$ are

$$F_1 = j_{ac} \cos(\tau + \beta), \quad F_2 = (j_1 \sin \varphi_0 + 4 j_2 \sin 2\varphi_0)\varphi_1^2/2, \quad \dots \tag{20}$$

To ensure smallness of $\varphi_n$ and applicability of the perturbation theory, we impose the condition $\overline{\dot{\varphi}_n} = 0$ which means that corrections to the phase do not grow with time. We write this condition in the form

$$\overline{\left[ j_{dc}^{(n)} + F_n(\tau) \right]/\dot{\varphi}_0(\tau)} = 0, \quad n = 1, 2, \dots, \tag{21}$$

and calculate $j_{dc}^{(n)}$ from this equation.

In the limiting case of small second harmonic ($A \ll 1$), we combine the perturbation theory with respect to $A$ developed in Ref. [29] with the perturbation theory with feedback discussed above. It means that each $\varphi_n$ (with $n = 0, 1, 2, \dots$) becomes a series with respect to $A$. In further calculations, we do not keep terms of the order of $A^2$ and higher.[1] In particular,

$$\varphi_0(\tau) = f_0(\tau) + A f_1(\tau), \tag{22}$$

where $f_0$ is the solution of Eq. (11) with $j_{dc}^{(0)}$ in the rhs, and $f_1$ was obtained in Ref. [29]:

$$f_1(\tau) = \frac{2\nu^2 \ln \left( j_{dc}^{(0)} + j_1 \cos(\nu\tau - \alpha) \right)/j_1 - 2 j_{dc}^{(0)} \cos(\nu\tau - \alpha)}{j_{dc}^{(0)} + j_1 \cos(\nu\tau - \alpha)}. \tag{23}$$

---

[1] As we discuss later, higher orders with respect to $A$ do not lead to qualitatively new features.

Then we substitute $\varphi_0$ in the expression for $\varphi_1$. The condition that $\varphi_1$ does not grow with time [condition (21)] in the first order of perturbation theory with respect to both $A$ and $j_{ac}/j_1$ gives

$$\overline{\left[ j_{dc}^{(1)} + j_{ac} \cos(\tau + \beta) \right] \left[ \frac{1}{\dot{f}_0(\tau)} - A \frac{\dot{f}_1(\tau)}{\dot{f}_0^2(\tau)} \right]} = 0. \tag{24}$$

If $A = 0$, the second bracket is a Fourier series with only zeroth and first harmonics of the internal Josephson oscillations, i.e., const and $e^{\pm i \nu \tau}$. At the same time, the first bracket contains harmonics $\pm 1$ due to external irradiation. Current correction $j_{dc}^{(1)}$ is hence nonzero only in the case $\nu = \pm 1$ due to synchronization between the two brackets (resonance between external irradiation and internal oscillations). This corresponds to the first integer Shapiro steps (positive and negative) with amplitude [28]

$$\Delta j_{\pm 1} = j_1 j_{ac} / \sqrt{1 + j_1^2}. \tag{25}$$

In the case $A \neq 0$, expansion of the second bracket into Fourier series additionally contains all harmonics $e^{\pm i k \nu \tau}$ with $k > 1$. This corresponds to additional resonances $\nu = \pm 1/k$, at which current correction $j_{dc}^{(1)}$ is nonzero:

$$j_{dc}^{(1)} = \frac{A j_{ac} \nu}{j_{dc}^{(0)}} \sin(\pm \beta + k\alpha)(-1)^{k+1} \left( \frac{(j_{dc}^{(0)} - \nu)^{k-1}}{(k-1)j_1^{k-1}} - \frac{(j_{dc}^{(0)} - \nu)^{k+1}}{(k+1)j_1^{k+1}} \right). \tag{26}$$

The argument of the sine in Eq. (26) contains phase shift between external ac signal and internal Josephson oscillations. This shift can be arbitrary (formally, due to arbitrary value of the phase $\beta$), hence the sine in Eq. (26) varies between $-1$ and $1$, which corresponds to the step in the CVC. In the first order of the perturbation theory, the two-harmonic CPR thus produces the series of fractional Shapiro steps of type $\pm 1/k$ (with integer $k > 1$) with amplitudes

$$\Delta j_{\pm 1/k} = 2 \frac{A j_{ac}}{\sqrt{k^2 j_1^2 + 1}} \left( \frac{(\sqrt{k^2 j_1^2 + 1} - 1)^{k-1}}{(k-1)(k j_1)^{k-1}} - \frac{(\sqrt{k^2 j_1^2 + 1} - 1)^{k+1}}{(k+1)(k j_1)^{k+1}} \right). \tag{27}$$

In particular, if $k = 2$ Eq. (27) reproduces formula (72) from Ref. [29] up to the redefinition of variables. At arbitrary $k > 1$, this result is equivalent to Eq. (35) from Ref. [16].

This series of steps is illustrated in Fig. 1(b). All these steps appear in the same order of the perturbation theory, and in the limit $k \to \infty$ the amplitudes of the steps decrease as

$$\Delta j_{\pm 1/k} \simeq 4 A j_{ac} e^{-1/j_1} (j_1 + 1)/k^3 j_1^2, \qquad k \gg \max(1, 1/j_1). \tag{28}$$

Are there fractional steps besides the $\pm 1/k$ series? In order to study this we consider the second order of perturbation theory with respect to $j_{ac}/j_1$ in the case $\nu \neq \pm 1/k$. The current correction $j_{dc}^{(1)}$ is hence equal to zero. Using the expression for $F_2$, we write the condition that $\varphi_2$ does not grow with time [condition (21)]:

$$\overline{\left[ j_{dc}^{(2)} + (j_1/2)\varphi_1^2 \sin \varphi_0 + 2 j_2 \varphi_1^2 \sin 2\varphi_0 \right] / \dot{\varphi}_0(\tau)} = 0. \tag{29}$$

Fourier series of $\varphi_1^2$ contains harmonics $e^{\pm i(2 - m\nu)\tau}$ with $m \in \mathbb{Z}$. Product of these terms with integer Fourier harmonics coming from functions of $\varphi_0$ contains the same harmonics with shifted $m$ which leads to nonzero result after averaging over $\tau$ in the case $\nu = 2/m$.

Substitution $m = \pm 1$ gives amplitude of the second integer Shapiro step which is

$$\Delta j_{\pm 2} = j_1 j_{ac}^2 / 4 \tag{30}$$

in the main order with respect to $A$ and exists even in the single-harmonic case ($A = 0$) [28]. At the same time, in the case of $m \neq \pm 1$ and $A = 0$, averaged terms in Eq. (29), proportional to $j_{ac}^2$, sum up to zero, hence $j_{dc}^{(2)} = 0$. This result is consistent with fact that the single-harmonic CPR produces only integer Shapiro steps. In the case $A \neq 0$, current correction $j_{dc}^{(2)}$ turns out to be nonzero at $m \neq \pm 1$ and, hence, fractional Shapiro steps of type $\pm 2/k$ with $k > 2$ appear. All these steps appear in the same order of the perturbation theory, and their amplitudes are

$$\Delta j_{\pm 2/k} \propto A j_{ac}^2. \tag{31}$$

Dependence on $A$ emphasizes that fractional steps appear due to the presence of the second Josephson harmonic in the CPR.

Analyzing the harmonic structure of Eq. (21) in higher orders of perturbation theory with respect to $j_{ac}/j_1$, we come to conclusion that if the CPR includes two harmonics, all fractional Shapiro steps appear, and in the limiting case $A \ll 1, j_{ac} \ll j_1$ amplitudes of fractional step $\pm n/k$ are

$$\Delta j_{\pm n/k} \propto A j_{ac}^n \tag{32}$$

in the main order of perturbation theory (while amplitudes of integer steps are $\Delta j_{\pm n} \propto j_{ac}^n$ ). The largest step sizes correspond to the "main" $\pm 1/k$ series [see Eq. (27) for exact result].

In the above discussion of fractional Shapiro steps, we have considered all orders of the perturbation theory with respect to $j_{ac}/j_1$ and only linear one with respect to $A$. The reason is that keeping terms of the order of $A^2$ and higher in Eq. (22) does not lead to the appearance of new Shapiro steps, while only producing corrections to amplitudes of the previously found steps. The point is that all possible Fourier harmonics in $1/\dot{\varphi}_0(\tau)$ in Eq. (21), and hence in $F_n(\tau)$ too, are already generated by the first order with respect to $A$.

## 3.2 Limit of large dc current ($j_{dc} \gg j_s$)

In this section we apply the perturbation theory with feedback in the limiting case of large dc current [30], defined by conditions

$$j_1/j_{dc} \ll 1, \quad j_2/j_{dc} \ll 1. \tag{33}$$

This implies that the dc component of the current is large compared to the supercurrent, $j_{dc} \gg j_s$. Compared to the previous limiting case, condition $A \ll 1$ is not required.

The perturbation theory with feedback in this limiting case is a modification of the approach employed in Sec. 3.1. It starts with expansions (16) and (17), however, small parameters are now given by Eq. (33).

In the zeroth order of the perturbation theory Eq. (15) becomes

$$\dot{\varphi}_0 = j_{dc}^{(0)} + j_{ac} \cos(\tau + \beta). \tag{34}$$

Its straightforward solution is

$$\varphi_0 = \Delta\varphi + j_{dc}^{(0)} \tau + j_{ac} \sin(\tau + \beta), \tag{35}$$

where $\Delta\varphi$ is the initial value of $\varphi_0$ and can be arbitrary. Phase shift $\beta$ is then redundant and we assume it equal to zero.

As we will see below, fractional Shapiro steps of the most general form $\pm n/k$ appear in this limiting case. According to Eq. (10), this must correspond to

$$j_{dc}^{(0)} = \pm n/k. \tag{36}$$

In the first order of the perturbation theory, expanding $\sin \varphi_0$ and $\sin 2\varphi_0$ into the Fourier series with respect to $\tau$, we obtain

$$\dot{\varphi}_1 = j_{\text{dc}}^{(1)} - j_1 \sum_{m=-\infty}^{\infty} J_m(j_{\text{ac}}) \sin\left((j_{\text{dc}}^{(0)} + m)\tau + \Delta\varphi\right) - j_2 \sum_{m=-\infty}^{\infty} J_m(2j_{\text{ac}}) \sin\left((2j_{\text{dc}}^{(0)} + m)\tau + 2\Delta\varphi\right), \tag{37}$$

where $J_m(z)$ are the Bessel functions of the first kind. This formula implies only integer ($j_{\text{dc}}^{(0)} = \pm n$) and half-integer ($j_{\text{dc}}^{(0)} = \pm n/2$) Shapiro steps. Their amplitudes are

$$\Delta j_{\pm n} = 2 \max_{\Delta\varphi} \left|j_1 J_n(j_{\text{ac}}) \sin \Delta\varphi + j_2 J_{2n}(2j_{\text{ac}}) \sin 2\Delta\varphi\right|, \tag{38}$$

$$\Delta j_{\pm n/2} = 2j_2 \left|J_n(2j_{\text{ac}})\right|, \tag{39}$$

respectively. The occurrence of trivial steps is directly caused only by the presence of the corresponding Josephson harmonic in the CPR. Equation (38) at arbitrary $n$ and Eq. (39) at $n = 1$ reproduce expressions obtained in Ref. [30] (in the $\beta = 0$ limit).

In order to find nontrivial fractional Shapiro steps (with $k > 2$), we assume $j_{\text{dc}}^{(0)} \neq \pm n$ and $\pm n/2$, which leads to $j_{\text{dc}}^{(1)} = 0$, and investigate the second order of the perturbation theory:

$$\dot{\varphi}_2 = j_{\text{dc}}^{(2)} - (j_1 \cos \varphi_0 + 2j_2 \cos 2\varphi_0)\varphi_1. \tag{40}$$

The Fourier series coming from $\varphi_1$ and functions of $\varphi_0$ contain harmonics $e^{\pm i(j_{\text{dc}}^{(0)} - m)\tau}$ and $e^{\pm i(2j_{\text{dc}}^{(0)} - m)\tau}$ with $m \in \mathbb{Z}$. Products of these terms generate harmonics with $(3j_{\text{dc}}^{(0)} - m)\tau$ and $(4j_{\text{dc}}^{(0)} - m)\tau$ in the argument of exponent, which may lead to nonzero result after averaging over $\tau$ in the cases $j_{\text{dc}}^{(0)} = m/3$ and $j_{\text{dc}}^{(0)} = m/4$.

If $j_{\text{dc}}^{(0)} = \pm n/3$, the condition that $\varphi_2$ does not grow with time gives

$$j_{\text{dc}}^{(2)} = \frac{3}{2} j_1 j_2 \cos(3\Delta\varphi) \sum_{m=-\infty}^{\infty} \frac{J_m(j_{\text{ac}}) J_{-m\mp n}(2j_{\text{ac}})}{\pm n + 3m}. \tag{41}$$

Due to arbitrary value of the initial phase $\Delta\varphi$, the cosine in Eq. (41) varies between $-1$ and $1$, which corresponds to the step in the CVC. In the second order of the perturbation theory, the two-harmonic CPR thus produces the series of fractional Shapiro steps of type $\pm n/3$ with amplitudes

$$\Delta j_{\pm n/3} = 3j_1 j_2 \left|\sum_{m=-\infty}^{\infty} \frac{J_m(j_{\text{ac}}) J_{-n-m}(2j_{\text{ac}})}{n + 3m}\right|. \tag{42}$$

In the case $j_{\text{dc}}^{(0)} = \pm n/4$, the condition that $\varphi_2$ does not grow with time gives

$$j_{\text{dc}}^{(2)} = 2j_2^2 \cos(4\Delta\varphi) \sum_{m=-\infty}^{\infty} \frac{J_m(2j_{\text{ac}}) J_{-m\mp n}(2j_{\text{ac}})}{\pm n + 2m}. \tag{43}$$

However, the sum in Eq. (43) is equal to zero, therefore, fractional Shapiro steps of type $\pm n/4$ do not appear in the second order of the perturbation theory. This is not unexpected. Fourier series that consists of harmonics $e^{\pm(4j_{\text{dc}}^{(0)} - m)\tau}$ in Eq. (40) does not contain $j_1$ and hence does not depend on the presence of the first Josephson harmonic in the CPR. If the series in Eq. (43) was nonzero, we could assume $j_1 = 0$, replace $2\varphi_0$ with $\varphi_0$ and obtain fractional half-integer Shapiro steps produced by the single-harmonic CPR. Vanishing of Eq. (43) emphasizes the fact that nontrivial fractional Shapiro steps appear only due to the interplay of different Josephson harmonics.

Are there nontrivial fractional steps beside the $\pm n/3$ series? In order to study this we consider higher orders of the perturbation theory with respect to $j_{1,2}/j_{\mathrm{dc}}$ in the case $j_{\mathrm{dc}}^{(0)} \neq \pm n/k$ with $k = 1, 2$, and 3. The current correction $j_{\mathrm{dc}}^{(2)}$ is hence equal to zero. In the third order of the perturbation theory:

$$\dot{\varphi}_3 = j_{\mathrm{dc}}^{(3)} - (j_1 \cos \varphi_0 + 2j_2 \cos 2\varphi_0)\varphi_2 + (j_1 \cos \varphi_0 + 4j_2 \cos 2\varphi_0)\varphi_1^2/2. \tag{44}$$

We proceed in the same manner as before, considering harmonics generated in the rhs of this expression.

As a result, in the third order of the perturbation theory we find series of fractional steps of types $\pm n/4$ and $\pm n/5$ with amplitudes

$$\Delta j_{\pm n/4} = 8j_1^2 j_2 \left| \sum_{m=-\infty}^{\infty} \sum_{l=-\infty}^{\infty} \frac{J_m(j_{\mathrm{ac}})J_l(j_{\mathrm{ac}})J_{-n-m-l}(2j_{\mathrm{ac}})}{(n+4m)(n+4l)} \right|, \tag{45}$$

$$\Delta j_{\pm n/5} = \frac{25}{4} j_1 j_2^2 \left| \sum_{m=-\infty}^{\infty} \sum_{l=-\infty}^{\infty} \frac{J_m(2j_{\mathrm{ac}})J_l(2j_{\mathrm{ac}})J_{-n-m-l}(j_{\mathrm{ac}})}{(2n+5m)(2n+5l)} \right|. \tag{46}$$

It is important to note that fractional Shapiro steps of type $\pm n/4$ appear due to product of the first and second harmonics, and fractional steps of type $\pm n/6$ do not appear, because in this order of the perturbation theory they cannot be produced by the interplay of Josephson harmonics.

According to the results above, we find a certain pattern that the amplitude of the fractional Shapiro steps of type $\pm n/k$ is

$$\Delta j_{\pm n/k} \propto j_1^\alpha j_2^\beta \quad \text{with} \quad \alpha + 2\beta = k \tag{47}$$

(while $\alpha + \beta$ is the order of the perturbation theory which should be minimized to get the main contribution to the steps size). At the same time, for nontrivial Shapiro steps both $\alpha$ and $\beta$ should be nonzero, because otherwise we would have a contribution to the steps size generated by a single Josephson harmonic, while we know that a single-harmonic CPR only produces trivial steps. As a result,

$$\beta = \left[ \frac{k-1}{2} \right]. \tag{48}$$

In the above derivation we have explicitly checked this pattern for $k$ from 1 to 5. Analyzing the harmonic structure of $\dot{\varphi}_n(\tau)$ in higher orders of perturbation theory with respect to $j_{1,2}/j_{\mathrm{dc}}$, we conclude that if the CPR contains the two harmonics, then all fractional Shapiro steps appear. According to Eqs. (47) and (48), fractional steps of types $\pm n/2k$ and $\pm n/(2k+1)$ appear in the $(k+1)$th order of perturbation theory with respect to $j_{1,2}/j_{\mathrm{dc}}$, and their amplitudes are

$$\Delta j_{\pm n/2k} \propto j_1^2 j_2^{k-1}, \qquad \Delta j_{\pm n/(2k+1)} \propto j_1 j_2^k. \tag{49}$$

## 3.3 Overlap of the two limiting cases

The conditions of the limiting cases considered in Secs. 3.1 and 3.2 are compatible if $j_{\mathrm{ac}}, j_2 \ll j_1 \ll j_{\mathrm{dc}}$. So, in this overlapping regime, we can compare the results of the above limiting cases. Approaching this regime from the limiting case of small amplitude of the ac current and small amplitude of the second Josephson harmonic (Sec. 3.1), we need to additionally assume $j_1 \ll j_{\mathrm{dc}}$ [equivalent to $kj_1 \ll 1$ in Eq. (27)]. Equation (27) then yields

$$\Delta j_{\pm 1/k} = j_1^{k-2} j_2 j_{\mathrm{ac}}/2^{k-2}(k-1). \tag{50}$$

Alternatively, we may approach the overlapping regime starting from the limiting case of large dc current (Sec. 3.2), additionally assuming $j_{ac}, j_2 \ll j_1$. Substituting $n = 1$, we then see that Eqs. (39), (42), and (45) reproduce Eq. (50) with $k = 2$, 3, and 4, respectively.

At the same time, the results of the two limiting cases differ from each other in the case of $\pm 1/k$ Shapiro steps with $k > 4$ [see Eq. (50) vs Eq. (49)]. Actually, they are both legitimate contributions to the amplitudes of the steps; however, their relative importance depends on an additional parameter $j_1^2/j_2$. In Sec. 3.2, we did not assume any special limit with respect to this parameter, so we minimized $\alpha + \beta$ in Eq. (47) in order to minimize the overall power of $j_1^\alpha j_2^\beta$, obtaining Eq. (48). However, if we assume $j_1^2 \gg j_2$, the main contribution stems from minimizing $\beta$, which yields $\beta = 1$. The dominant contribution to the amplitude of the fractional steps of $\pm 1/k$ type is then given by Eq. (50). In the opposite case, $j_1^2 \ll j_2$, Eq. (48) is valid and the main result is given by Eq. (49) with $n = 1$.

Generalization of the above reasoning to arbitrary nontrivial fractions $\pm n/k$ (with $k \geq 3$) is straightforward. In the case $j_1^2 \gg j_2$, the amplitude of the step is given by Eq. (32) [instead of Eq. (50)], which can be refined as

$$\Delta j_{\pm n/k} \propto j_1^{k-2} j_2 j_{ac}^n. \tag{51}$$

In the opposite case, $j_1^2 \ll j_2$, the main result is given by Eq. (49), which can be refined as

$$\Delta j_{\pm n/2k} \propto j_1^2 j_2^{k-1} j_{ac}^n, \qquad \Delta j_{\pm n/(2k+1)} \propto j_1 j_2^k j_{ac}^n. \tag{52}$$

As before, the differences arise at $k > 4$. The reason is that at such values of $k$, Eq. (47) has several solutions with respect to nonzero $\alpha$ and $\beta$.

## 4   More harmonics in the current-phase relation

In this section, we consider the CPR with three Josephson harmonics and then discuss generalization to the case of more harmonics. In the limiting case of small amplitude of the higher Josephson harmonics and ac current, considered in Sec. 3.1, presence of the third harmonic leads to the complication of Eq. (18) and expressions for $F_n(\tau)$ with $n > 1$. However, similarly to the terms $\propto A^2$, it does not lead to the appearance of new Shapiro steps. The point is that all possible Fourier harmonics in $1/\dot{\varphi}_0(\tau)$ in Eq. (21), and hence in $F_n(\tau)$ too, are already generated by the presence of the second Josephson harmonic in the CPR. At the same time, physically, the condition $A \ll 1$ can be due to the small transparency $T \ll 1$ of the Josephson junction. If this is indeed so, then the amplitude of the $n$th Josephson harmonic is proportional to $T^n$, hence the normalized amplitude of the third harmonic is $\sim A^2$. Along the same lines as in the end of Sec. 3.1, we conclude that the presence of the third Josephson harmonic (as well as higher ones) does not affect the amplitude of the nontrivial fractional Shapiro steps in the main order of the perturbation theory.

In the limiting case of large dc current, considered in Sec. 3.2, CPR with three Josephson harmonics produce all fractional steps in lower orders of the perturbation theory, compared to the case of only two Josephson harmonics. Similarly to the case of two harmonics, amplitude of the fractional Shapiro steps of type $\pm n/k$ is proportional to $j_1^\alpha j_2^\beta j_3^\gamma$ with $\alpha + 2\beta + 3\gamma = k$ (while $\alpha + \beta + \gamma$ is the order of the perturbation theory which should be minimized to get the main contribution to the steps size). At the same time, for nontrivial Shapiro steps two out of three $\alpha$, $\beta$, and $\gamma$ should be nonzero, because otherwise we would have a vanishing contribution to the steps size generated by a single Josephson harmonic. (The above consideration is straightforwardly generalized to the case of more Josephson harmonics.)

At the same time, fractional Shapiro steps of type $\pm n/3$ now become trivial and appear in the first order of the perturbation theory. Applying the rule described above, we conclude that

steps of type $\pm n/4$ and $\pm n/5$ appear in the second order of the perturbation theory due to the interplay of the third Josephson harmonic with the first and the second one, respectively. Amplitudes of these steps are $\propto j_1 j_3$ and $\propto j_2 j_3$, respectively.

# 5   Josephson diode effect

In this section, we discuss the manifestation of the Josephson diode effect (JDE) in fractional Shapiro steps. The diode effect refers to the dependence of a junction's properties on the direction of the current. We consider a modification of the CPR in Eq. (5) by a phase shift $\widetilde{\phi}$ between the two Josephson harmonics, so that the dimensionless CPR becomes

$$j_s(\varphi) = j_1 \sin\varphi + j_2 \sin(2\varphi - \widetilde{\phi}). \tag{53}$$

This form of the CPR effectively arises in different physical systems such as combined $0$-$\pi$ JJs in magnetic field [31], JJs between exotic superconductors with broken time-reversal symmetry [32], asymmetric SQUIDs with higher Josephson harmonics [29,33], and SQUIDs with three sinusoidal JJs in the loop [34].

If $\widetilde{\phi} \neq 0 \bmod \pi$, our numerical calculations demonstrate asymmetry between amplitudes of positive and negative fractional steps. Analytically, the presence of $\widetilde{\phi}$ slightly changes current corrections in Eqs. (26) and (41): $\pm\beta + k\alpha + \widetilde{\phi}$ instead of $\pm\beta + k\alpha$ in the argument of sine and $3\Delta\varphi - \widetilde{\phi}$ instead of $3\Delta\varphi$ in the argument of cosine, respectively. This means, that fractional Shapiro steps remain symmetric in the main order of the perturbation theory. In order to find asymmetry, higher orders of the perturbation theory are required.

In the limiting case of small amplitude of the ac current and small amplitude of the second Josephson harmonic, we further develop the derivation of Sec. 3.1 and consider the second order with respect to $A$. To illustrate the results, we consider the main series of the fractional Shapiro steps. The corresponding asymmetric correction to the steps size can be written as

$$\Delta j_{\pm 1/k}^{(\mathrm{asym})} = \mp\gamma j_{\mathrm{ac}} A^2 \sin\widetilde{\phi}, \tag{54}$$

where the $\gamma$ factor is positive and depends only on $j_{\mathrm{dc}}$ and $j_1$.

In the limiting case of large dc current we further develop the derivation of Sec. 3.2. Considering the next order with respect to $j_{1,2}/j_{\mathrm{dc}}$, we find asymmetric corrections to the amplitudes of fractional steps, which are

$$\Delta j_{\pm n/2k}^{(\mathrm{asym})} = \pm\left(\gamma_1 j_1^4 j_2^{k-2} + \gamma_2 j_1^2 j_2^k\right)\sin\widetilde{\phi}, \tag{55}$$

$$\Delta j_{\pm n/(2k+1)}^{(\mathrm{asym})} = \pm\left(\gamma_3 j_1^3 j_2^{k-1} + \gamma_4 j_1^1 j_2^{k+1}\right)\sin\widetilde{\phi}, \tag{56}$$

where $\gamma_1$, $\gamma_2$, $\gamma_3$, and $\gamma_4$ depend only on $j_{\mathrm{dc}}$ and $j_{\mathrm{ac}}$.

# 6   Discussion

Now we comment on the difference between the RSJ model that we have studied and its possible modifications.

A modification which is often considered in the literature corresponds to a different driving regime: instead of the current-driven case, one can assume the voltage-driven case. In this regime, not the current but the voltage is fixed, $V(t) = V_{\mathrm{dc}} + V_{\mathrm{ac}} \cos\omega t$. The phase $\varphi(t)$ is then found not from the RSJ equation (1) but directly from the Josephson relation (2). The result determines the current $I_s(\varphi(t))$ that demonstrates the Shapiro features (spikes) in the $\overline{I_s}(V_{\mathrm{dc}})$

dependence at $V_{dc}$ given by the same relation as Eq. (3). In the case of only one harmonic in CPR, only integer Shapiro spikes arise [6, 35]. Higher harmonics in the CPR lead to fractions with $k > 1$, and the denominators directly correspond to the harmonic number [35]. This means that irreducible Shapiro fractions with a given $k$ appear only due to the presence of the harmonics with numbers $m$ that are multiples of $k$ in the CPR. As a result, a CPR with only two harmonics, Eq. (5), generates only integer and half-integer Shapiro features.

Returning to the current-driven regime, we now focus on the sinusoidal CPR with $I_s(\varphi) = I_c \sin\varphi$. As we have already mentioned, in this case, the RSJ model (1) produces only integer steps [6, 7, 20–22]. At the same time, this model can be considered as the overdamped limit of the more general Resistively and Capacitively Shunted Junction (RCSJ) model [6, 7]. The RCSJ model differs from Eq. (1) by additional term $(\hbar C/2e)d^2\varphi/dt^2$ in the left-hand side (where $C$ is the capacitance). This modification leads to the generation of all subharmonic steps and to the devil's staircase structure of the CVC even with in the case of the sinusoidal CPR [36–38].

Finally, we comment on experimental relevance of a CPR with only two Josephson harmonics ($m = 1, 2$). Such CPR can be a good approximation in JJs close to the tunneling limit. Theoretically, higher Josephson harmonics contain additional powers of small junction transparency. In a recent experiment [39], the second Josephson harmonic was shown to be the leading correction to the first one in tunnel junctions. Alternatively, the second harmonic may become important if the first one is suppressed due to additional physical mechanisms, e.g., near the 0-$\pi$ transition [40–44].

# 7  Conclusions

In the framework of the RSJ model (9), we have theoretically investigated a Josephson junction with the CPR (5) containing two Josephson harmonics. We demonstrate that due to the presence of the second Josephson harmonic, in the current-driven regime, not only integer and half-integer but also all fractional (subharmonic) steps appear. We employ a combination of perturbative methods to find analytically amplitudes of the fractional Shapiro steps (generally given by $\bar{v} = \pm n/k$ which we assume to be irreducible fractions).

In the limiting case of small amplitude of the ac current and small amplitude of the second Josephson harmonic ($j_{ac} \ll j_1$, $A = j_2/j_1 \ll 1$), in the first order of the perturbation theory, we reproduce the result of Ref. [16] for the amplitudes of fractional steps of type $\pm 1/k$ with $k > 1$. The amplitudes of these steps are $\propto A j_{ac}$ [see Eq. (27)]. In the next order of the perturbation theory with respect to $j_{ac}/j_1$, we find fractional steps of type $\pm 2/k$ with $k > 2$; their amplitudes are $\propto A j_{ac}^2$. The structure of the perturbation theory suggests that the $n$th order reveals fractional steps of type $\pm n/k$ with amplitudes $\propto A j_{ac}^n$. At the same time, higher orders of the perturbation theory with respect to $A$ only provide small corrections to the above results.

In the limiting case of large dc current, we find amplitudes of nontrivial fractional steps of type $\pm n/3$ in the second order of the perturbation theory with respect to $j_1/j_{dc} \ll 1$ and $j_2/j_{dc} \ll 1$. Their amplitudes are $\propto j_1 j_2$ [see Eq. (42)]. We also find amplitudes of fractional steps of types $\pm n/4$ and $\pm n/5$ in the third order of the perturbation theory; their amplitudes are $\propto j_1^2 j_2$ and $\propto j_1 j_2^2$, respectively [see Eqs. (45) and (46)]. The structure of the perturbation theory suggests that the $(k + 1)$th order reveals fractional steps of types $\pm n/2k$ and $\pm n/(2k + 1)$ with amplitudes $\propto j_1^2 j_2^{k-1}$ and $\propto j_1 j_2^k$, respectively. Importantly, nontrivial fractional steps appear due to product of different Josephson harmonics and do not arise in the single-harmonic case.

Additionally, we consider the CPR with a phase shift between the two Josephson harmonics, Eq. (53), which leads to the Josephson diode effect. In the case of fractional steps, this is

manifested in asymmetry between the $n/k$ and $-n/k$ steps. In the limiting cases, we find the corresponding asymmetries which turn out to be small compared to the amplitudes of the steps. The results are presented in Eqs. (54)-(56).

## Acknowledgements

We thank Yu. M. Shukrinov for useful discussions.

**Funding information** The work was supported by the Ministry of Science and Higher Education of the Russian Federation (state assignment for the Landau Institute for Theoretical Physics).

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
