# Peer review of "All fractional Shapiro steps in the RSJ model with two Josephson harmonics"

_SciPost Physics_

## Round 1 · Referee Report · Anonymous (Referee 1) · 2025-7-29

Report
In this manuscript, the authors derive a very interesting and intriguing result: the presence of just two harmonics in the current-phase relation of a Josephson junction is enough to generate all possible fractional Shapiro steps in the current-voltage characteristic of this junction with a resistive shunt, when subject to a combination of dc and monochromatic ac drives. In addition to the sheer mathematical beauty of this result, it may have rather important implications for metrology, since Shapiro steps are used as a basis for the voltage standard, and having multiple harmonics in the current-phase relation is rather common for many Josephson junctions. This work clearly opens a new direction in this field, with a potential for future studies, theoretical and especially experimental. The authors approach the problem perturbatively from two sides, and find an agreement between the two. The paper itself is well written, and the theoretical approach is quite rigorous, so in my opinion the paper meets the standards of SciPost Physics.
Still, before recommending the paper for publication, I would like the authors to clarify a couple of points about their approach.
-
The authors obtain their results by extending the procedure proposed in Ref. [28], the perturbation theory with feedback. While the procedure itself is clearly described, its meaning is not totally clear to me (even upon looking up Ref.[28]). In Eq.(9), j_dc serves as an external control parameter which can take any value. Eq.(9) is dissipative, and it is supposed to have finite solutions for any value of j_dc. What is the meaning of the series (17) then? Do I understand correctly that for some values of j_dc these solutions can be constructed perturbatively, while for some other values they cannot, and if we happen to choose a "bad" value of j_dc, then we are obliged to push it away from the "bad" region by introducing the corresponding correction? In other words, does it mean that the solution can be constructed perturbatively only outside the Shapiro steps?
-
In the discussion section, the authors note that under a voltage bias, a junction with two harmonics will exhibit only integer and half-integer Shapiro steps. To me this looks puzzling, since one can continuously switch between the voltage-bias and current-bias protocols by introducing a load resistor, and all such situations should be equivalent according to Thévenin's and Norton's theorems. How can one have all fractional Shapiro steps in one protocol, and only integer and half-integer in the other?
In addition to these two points, I noticed a couple of misprints.
In Eq.(1), R is not the normal-state resistance of the junction. The junction is in the superconducting state, and R represents any additional mechanism of dissipative charge transfer, acting simultaneously (in parallel) with the non-dissipative Cooper pair tunnelling (e.g., tunnelling of Bogolyubov quasiparticles).
In Eq.(5), the second coefficient should be I_2.
Recommendation
Ask for minor revision

Author: Yakov Fominov on 2025-11-04 [id 5989]
(in reply to Report 3 on 2025-08-15)Dear Referee, thank you for your report and the comments provided. Below, we present our response to the points you raised.
According to you suggestion, we have performed stability analysis of our perturbative solutions. The results demonstrate that a stable solution always exists. In the resubmitted manuscript, we will add a new Sec. 3.4 devoted to this issue.
We have run our manuscript through a grammar checking program. As a result, a number of articles will be corrected in the resubmitted manuscript.

---

## Round 1 · Referee Report · Anonymous (Referee 2) · 2025-8-5

Report
Before evaluating the results of the calculations, which are indeed non-trivial, I would like to discuss a couple of key points underlying the approach used.
1. A phase shift between the external ac signal and the internal Josephson oscillations is introduced into the calculations (or rather, artificially maintained). However, the phase-locking effect underlying the appearance of the Shapiro steps corresponds (‘by definition’) to the absence of such a shift. The assumption made that this shift can be arbitrary is most likely erroneous. It is well known, for example, that by means of an external ac signal it is possible to synchronize even all phases in a stack of Josephson junctions to obtain the ‘giant Shapiro steps’.
2. The phase shift between two Josephson harmonics, introduced in Section 5, which would seem to break the CPT symmetry in the absence of an external magnetic field, appears to be equally unlikely.
Thus, the authors of the manuscript must first somehow justify the possibility of the above assumptions before discussing the implications of the results obtained.
Recommendation
Ask for major revision

---

## Round 1 · Referee Report · Anonymous (Referee 3) · 2025-8-15

Strengths
Report
Requested changes
- Perform stability analysis of the perturbative solutions.
- The manuscript, while being quite readable, misses a lot of articles. I would like to ask the authors to e.g. run it through a grammar checking program.
Recommendation
Ask for minor revision

---

## Editorial Decision

unknown